# A catalogue of biochemically diverse CRISPR-Cas9 orthologs

Giedrius Gasiunas [1,8], Joshua K. Young [2,8✉], Tautvydas Karvelis [3], Darius Kazlauskas [3], Tomas Urbaitis[1,3], Monika Jasnauskaite[1], Mantvyda M. Grusyte[1], Sushmitha Paulraj[2], Po-Hao Wang [2,6], Zhenglin Hou[2], Shane K. Dooley[4], Mark Cigan[2,7], Clara Alarcon[2], N. Doane Chilcoat[2], Greta Bigelyte[3], Jennifer L. Curcuru[5], Megumu Mabuchi[5], Zhiyi Sun [5], Ryan T. Fuchs[5], Ezra Schildkraut [5], Peter R. Weigele [5], William E. Jack[5], G. Brett Robb [5✉], Česlovas Venclovas[3] & Virginijus Siksnys [1,3✉]

Bacterial Cas9 nucleases from type II CRISPR-Cas antiviral defence systems have been repurposed as genome editing tools. Although these proteins are found in many microbes, only a handful of variants are used for these applications. Here, we use bioinformatic and biochemical analyses to explore this largely uncharacterized diversity. We apply cell-free biochemical screens to assess the protospacer adjacent motif (PAM) and guide RNA (gRNA) requirements of 79 Cas9 proteins, thus identifying at least 7 distinct gRNA classes and 50 different PAM sequence requirements. PAM recognition spans the entire spectrum of T-, A-, C-, and G-rich nucleotides, from single nucleotide recognition to sequence strings longer than 4 nucleotides. Characterization of a subset of Cas9 orthologs using purified components reveals additional biochemical diversity, including both narrow and broad ranges of temperature dependence, staggered-end DNA target cleavage, and a requirement for long stretches of homology between gRNA and DNA target. Our results expand the available toolset of RNA-programmable CRISPR-associated nucleases.

[1] CasZyme, Vilnius LT-10257, Lithuania. [2] Department of Molecular Engineering, Corteva Agriscience™, Johnston, IA 50131, USA. [3] Institute of Biotechnology, Vilnius University, Vilnius LT-10257, Lithuania. [4] Department of Agricultural and Biosystems Engineering, Iowa State University, Ames, IA 50011, USA. [5] New England Biolabs, Ipswich, MA 01938, USA. [6] Present address: Inari Agriculture, West Lafayette, IN 47906, USA. [7] Present address: Genus plc, Deforest, WI 53532, USA. [8] These authors contributed equally: Giedrius Gasiunas, Joshua K. Young. ✉email: josh.young@corteva.com; robb@neb.com; siksnys@ibt.lt

The Cas9 protein from type II CRISPR (clustered regularly interspaced short palindromic repeats)-Cas (CRISPR-associated) antiviral defense systems have been repurposed as a robust genome-editing tool (reviewed in refs. [1,2]). DNA target recognition is accomplished with small noncoding RNAs that through direct base pairing guide Cas9 to its DNA target site[3,4]. In addition to guide RNA (gRNA) recognition, a sequence motif, termed the protospacer adjacent motif (PAM), is required for the initiation of Cas9-guide RNA target binding and cleavage[3,4]. Easily reprogrammed to recognize different DNA sequences, it has been widely adopted for use in a multitude of applications to edit genomic DNA, modulate gene expression, visualize genetic loci, or detect targets in vitro[5–9]. To date, just a handful of variants are used for these applications[10–20] with the *Streptococcus pyogenes* (Spy) Cas9 being used most widely[2].

Since Cas9 can be programmed to target DNA sites by altering the spacer sequence of the gRNA, recognition of the PAM becomes a constraint that restricts the sequence space targetable by Cas9. This is further limited by the requirement for careful site selection to minimize off-target binding and cleavage based on the tolerance for mismatches in the gRNA–PAM–target complex[1,21]. This constraint becomes particularly evident in therapeutic applications where even rare genome alterations resulting from off-targets are undesirable or when targeting more structurally complex plant genomes[22,23]. Moreover, these restraints impact the use of Cas9 for homology-directed repair (HDR), template-free editing, base editing, or prime-editing applications, where the outcome is reliant on the proximity of the desired change to the target sequence[1]. Furthermore, the biochemical and physical characteristics of Cas9s routinely applied, producing predominantly blunt-end DNA target cleavage[3,4], slow substrate release[24], low frequency of recurrent target-site cleavage[25], gRNA exchangeability[26], temperature dependence[27], and size[28] may also be unfavorable for its varied applications. While Spy Cas9 targeting constraints are beginning to be addressed through structure-guided rational design[18,29] and directed evolution approaches[30–32], the diversity provided by naturally occurring orthologs may offer unique insight and opportunities for improvement of this powerful tool.

Here, we determine the gRNA and PAM requirements for 79 phylogenetically distinct Cas9s of various sizes without the need for protein purification or extensive computational analyses[33]. In doing so, we identify extraordinary diversity in Cas9 PAM and gRNA requirements. This extends the number of unique classes of gRNAs from four to seven and reveals T-, A-, C-, and G-rich PAM recognition that varies in length from one to more than four nucleotides. Interestingly, the analysis of the PAM interacting (PI) domain indicates that much of this variation is derived from just four related groups. Finally, additional biochemical studies reveal diversity that may further extend the application. This includes differences in temperature and spacer length requirements as well as variation in the pattern of double-stranded DNA (dsDNA) target cleavage.

## Results

**Cas9 ortholog selection.** To systematically sample diversity, 47 orthologs were chosen from most of the 10 major clades of a Cas9 evolutionary tree (Fig. 1a and Supplementary Data 1). Clades giving rise to previously characterized proteins that were active in eukaryotic cells were mined at a rate of ~20%, while all others were surveyed at a rate of ~10%. To enrich for proteins with robust biochemical activity and thermostability, an additional 32 orthologs were selected based on their physiochemical properties (e.g., predicted secondary structure and isoelectric point), classification as a type II-A subtype[26,34], and affiliation with a thermophilic host organism (Supplementary Data 2). Sequence length variation of our collection matched that found in naturally occurring orthologs and ranged from ~1000 to ~1600 residues with a bimodal distribution focused around sizes of ~1100 and ~1375 amino acids (Supplementary Fig. 1). Furthermore, amino acid sequence alignments of those selected showed extraordinary variation relative to each other and orthologs previously described to function as genome editing reagents[10–20], altogether, differing by as much as 93% (Supplementary Data 1).

**Guide RNA requirements.** In all instances, Cas9 gRNAs, the crRNA (CRISPR RNA) and tracrRNA (trans-activating CRISPR RNA), were identified near the *cas9* gene; however, spatial positioning, as well as the transcriptional orientation varied greatly among the systems characterized (Supplementary Fig. 2). In general, these features were conserved among orthologs belonging to a particular phylogenetic clade (Supplementary Fig. 2). Most CRISPR repeats were ~36-bp length; however, longer repeats (45–50 bp), associated with orthologs from clade X, were also identified (Supplementary Data 2). Computational analyses comparing co-variant models (CMs) based on sequence and secondary structure homology among the characterized tracrRNAs showed seven distinct clusters (Fig. 2). For some Cas9 orthologs, the tracrRNA self-clustered or demonstrated weak similarity to other CMs (Fig. 2). In these cases, it was not assigned to a particular group. In general, clusters were tightly associated with a particular Cas9 phylogenetic clade, although exceptions were noted (Fig. 2). Examination of the sgRNA modules (repeat: anti-repeat duplex, nexus, and 3′ hairpin-like folds)[35,36] was also typically conserved among related Cas9 proteins (Supplementary Fig. 3). For example, the sgRNA solutions for almost all members of clade IV resembled that belonging to Spy Cas9 and comprised a bulge in the repeat:anti-repeat duplex, a short nexus-like stem loop, and two hairpins followed by a poly-U sequence at the 3′ end[35,36]. Analogous structures were observed in the sgRNAs of Cas9 proteins belonging to clades VIII and X. However, in clade X sgRNAs, the repeat:anti-repeat duplexes were typically fully complementary and did not form repeat:anti-repeat bulges. Members of clade V contained the shortest sgRNAs, and reminiscent of the sgRNA from *Streptococcus aureus* (Sau) Cas9, these contained only two hairpins (nexus-like followed by a larger fold) following the repeat:anti-repeat duplex (Supplementary Fig. 3). In contrast, sgRNAs associated with clades III, VI, and IX orthologs displayed longer, more complex, and diverse structures. These included a variety of differences in stem length, presence of bulges, and different spacing between sgRNA modules (Supplementary Fig. 3). In addition, it was more difficult to reliably identify a Rho-independent-like terminator at the end of some tracrRNA encoding regions for these clades.

**PAM recognition by orthologous Cas9s.** To rapidly survey the target recognition properties of Cas9 orthologs, we employed a cell-free in vitro translation (IVT) method similar to that described previously (Fig. 1b)[33,37]. Since PAM recognition is dependent on the concentration of Cas9-guide RNA complex[19], crude IVT RNP mixtures were diluted ($10^1$–$10^3$ in tenfold increments) and tested for their ability to support cleavage when combined with a plasmid library containing a randomized PAM region adjacent to a Cas9 target site. The greatest dilution supporting cleavage activity was then used as a baseline for PAM recognition. To confirm the accuracy of our approach, Cas9 PAM recognition was examined using purified components, as described previously[19]. This was done for Spy, *S. thermophilus* CRISPR3 (Sth3), and *S. thermophilus* CRISPR3 (Sth1) Cas9s, whose PAM was determined previously[19] and for 11 orthologs

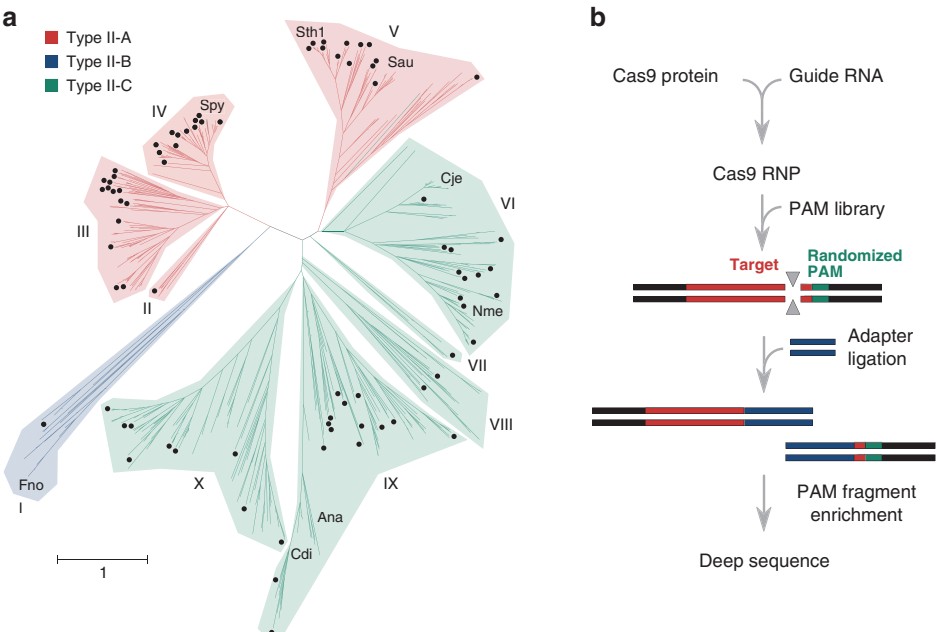

**Fig. 1 Cas9 diversity and characterization approach. a** Phylogenetic representation of the diversity provided by Cas9 orthologs. Type II-A, B, and C systems are color-coded, red, blue, and green, respectively. Distinct phylogenetic clades are numbered I–X. Those selected for the study are indicated with a black dot. Cas9s whose structure has been determined are also designated. **b** Biochemical approach used to directly capture target cleavage and assess protospacer adjacent motif (PAM) recognition. Experiments were assembled using Cas9 protein produced by IVT.

from our collection. As shown in Supplementary Fig. 4a, b, there was a nearly perfect agreement between the approaches. Additionally, to examine the propensity for PAM recognition to extend beyond position 7 (the length of randomization in our PAM library), the spacer targeting the PAM library was also shifted 5′ by 1, 2, or 3 nts for Cas9 orthologs that exhibited PAM preferences at positions 6 or 7 and lacked PAM requirements in the first, second or third positions. This permitted PAM identification to be extended to 8, 9, or 10 bp, respectively. Of the 20 Cas9s that were tested (Supplementary Data 2), only 6, all belonging to phylogenetic clade VI (Figs. 1a and 3), had PAM recognition that continued beyond the 7th position. Surprisingly, PAM preferences at the 8th position were always an A residue similar to the previously characterized *Brevibacillus laterosporus* (Blat)[19] and *Geobacillus stearothermophilus* (Geo)[16] Cas9 proteins. Altogether, we found that long PAMs extending beyond the 7th position were not widespread and only abundant in one family of orthologs belonging to clade VI.

The Cas9 orthologs characterized (Supplementary Data 2) with our IVT-based approach demonstrated significant divergence in PAM recognition. Indeed, we identified nucleases with previously undescribed PAM requirements that varied in composition both in sequence and length. Among these were proteins with PAM recognition that could be generally sub-divided into A-, T-, and C-rich PAM recognition in addition to the G-rich PAM typical of the Spy Cas9 protein (Fig. 3). PAMs composed of multiple residues of a single base pair, while present (e.g., Efa, Nme2, Rsp, Ssi, and Ssu), were rare but notably enriched in Clade IV (e.g Efa) to which Spy Cas9 belongs, and in Clade VII (Ssi, Ssu) (Fig. 1a, Fig. 3, and Supplementary Data 3). In general, Cas9s with composite PAM recognition containing at least two different base pairs were more abundant (Fig. 3). The length of PAM recognition also varied between 1 and 4 base pairs or more, with most orthologs exhibiting recognition at three or more positions. Additionally, many proteins exhibited seemingly degenerate PAM recognition. Typically, this resulted in a strong requirement for at least one base pair in combination with positions that accepted

more than one (typically two) base pairs (e.g., Lan, Mse, Nsa, and Sma2).

**Diversity and taxonomic distribution of Cas9 PAM interacting domains**. The extreme diversity of experimentally determined PAM sequence requirements prompted us to evaluate the sequence relationship of Cas9 PAM interacting (PI) domains. To do so, we extracted the PI regions from the characterized orthologs and used them as queries for iterative searches against non-redundant collections of microbial proteins (see "Methods"). In all, 9161 sequences having non-identical PI domains were found (Supplementary Data 4). Sequences were next clustered based on their pairwise similarity leading to the identification of ten clusters (Fig. 3). Clusters 1–4 were the largest and contained 93% of all sequences recovered, while clusters 7–10 were considerably smaller and were comprised of 4–37 sequences (Fig. 3 and Supplementary Data 4). Sequence searches with HHpred[38] showed that most clusters were distantly related to each other (Fig. 3 and Supplementary Fig. 5a–c), with an exception being cluster 10 that did not reveal significant similarity to any other group (Supplementary Fig. 5d). In general, PI domain similarity could be correlated with the major phylogenetic branches of the Cas9 tree (Figs. 1a and 3). For example, Cas9s belonging to clades II, III, and IV grouped into cluster 1 (Fig. 3). Additionally, phylogenetic analysis of clusters 1–6 also suggested that similar PI domains usually resulted in similar PAM recognition (Supplementary Fig. 6a, b, d); however, sequence diversity and length varied greatly even among members of the same group (Supplementary Fig. 6c). Closer examination of the Cas9s belonging to cluster 1 further highlighted that even within similar PI architectures sequence composition varied considerably with conservation being the lowest in the PI domain relative to the rest of the Cas9 protein (Supplementary Fig. 7).

Although clusters shared amino acid sequence similarity (Fig. 3), their taxonomic distribution differed. While PI domains from cluster 2 were mainly found in *Bacteroidetes* and *Alphaproteobacteria* (Supplementary Fig. 6b), cluster 3 was more

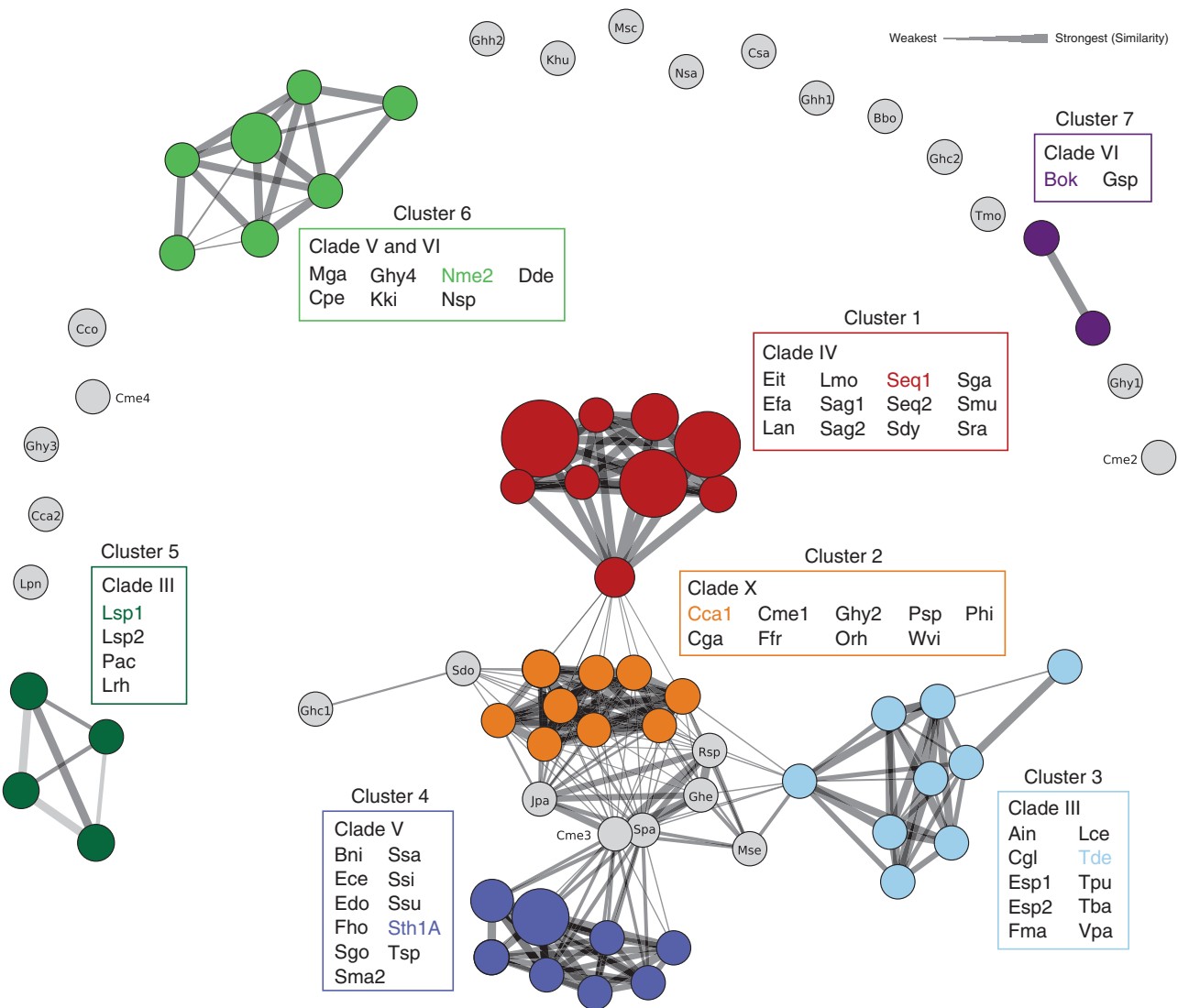

**Fig. 2 Cas9 tracrRNA sequence and secondary structure similarity.** Circles are scaled based on the number of sequences belonging to each covariance model (CM) and colored according to the designated cluster. The width of the connecting lines indicates the percentage of similarity or relatedness among CMs. Representative tracrRNAs from each cluster are indicated with the associated color. CMs not assigned to a cluster are in gray.

likely to come from *Betaproteobacteria*, *Epsilonproteobacteria*, and *Firmicutes* (*Bacilli* and *Clostridia*) (Supplementary Fig. 6c). Sequences from clusters 1 and 4 were usually found in *Firmicutes* (Supplementary Fig. 6a, d), while clusters 5 and 6 were specific to *Actinobacteria* and *Proteobacteria*, respectively (Supplementary Fig. 5e).

**Evaluation of Cas9 ortholog biochemical activity.** Fifty-two Cas9 orthologs from our collection were selected for additional characterization using purified components. Primary selection criteria included simple PAM recognition (≤3 bp) (where possible) while maintaining diversity in phylogenetic distribution and protein size. It was previously reported that Sau, Geo, Cje, and Nme2 Cas9 proteins require a spacer longer than 20 nt (which is optimal for Spy Cas9) to function robustly[11,15–17]. Therefore, we designed sgRNAs with two different spacer lengths, 20 and 24 nt, for each ortholog to initially gauge the influence of spacer length on Cas9 cleavage activity in vitro. Exceptions to this included Efa, Lpn, and Cme4, where a single spacer length of either 20 or 22 nt was tested. As shown in Supplementary Fig. 8, most orthologs worked best with a 20-nt spacer similar to Spy Cas9 when

evaluated across a panel of five different buffers; however, six orthologs, Cga, Cca1, Orh, Tmo, Nsa, and Ghh1 Cas9, required a spacer length of greater than 20 nt to effectively cut their DNA target. In all, 46 out of 52 produced dsDNA target cleavage activity greater than 25% under the conditions examined.

The thermal stability of 38 orthologs showing robust target cleavage was next predicted using nano differential scanning fluorimetry (nanoDSF). In all, 36 of 38 proteins showed a melting temperature of >37 °C confirming stability under standard in vitro enzymatic reaction conditions. Interestingly, five orthologs had melting temperatures >50 °C, suggesting thermo-stability (Supplementary Fig. 9). These included Cme2, Cme4, Ghy1, Esp1, and Nsa Cas9.

To corroborate nanoDSF predictions, DNA target cleavage was next measured in reactions at temperatures ranging from 10 °C to 68 °C. In all, Cas9 orthologs displayed a wide spectrum of temperature dependencies, including both narrow and broad ranges of activity (Fig. 4a). Consistent with thermal unfolding analysis, Cme2, Esp1, Nsa, Ain, Cme3, and Sth1A, were active at temperatures greater than 50 °C with Nsa, isolated from the deep-sea hydrothermal vent chimney bacterium, *Nitratifractor salsuginis*[39], remaining active at temperatures greater than 60 °C

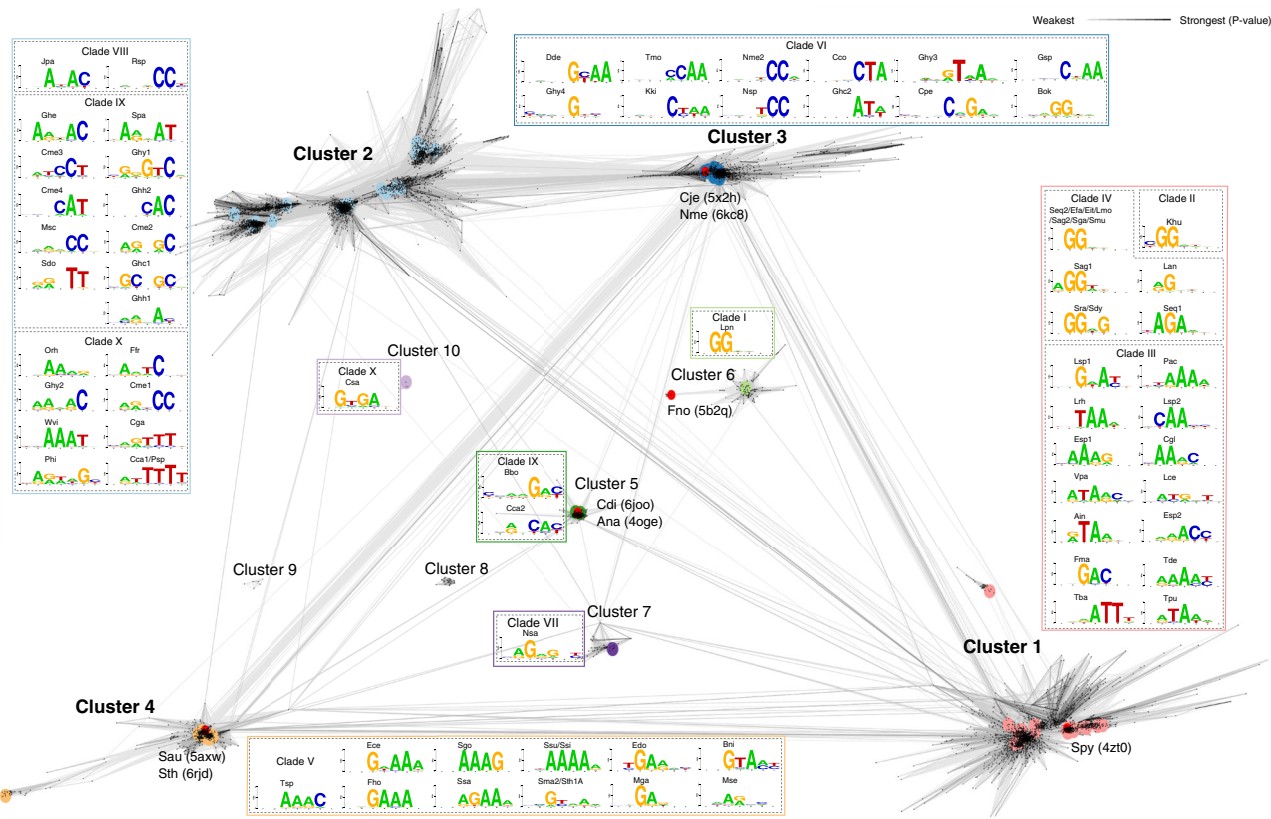

**Fig. 3 Cas9 protospacer adjacent motif (PAM) interacting (PI) domain similarity.** Cas9 PI domains clustered by their pairwise sequence similarity. Sequences were clustered using CLANS (BLAST option). Lines connect sequences with $P$ value $\leq$ 1e − 11. Line shading corresponds to $P$ values according to the scale in the top-right corner (light and long lines connect distantly related sequences). For details on how $P$ values are calculated, please see the "Methods" section. Major clusters are shown in bold. Cluster 1 was so named to emphasize that it contains the first experimentally characterized Cas9, Spy. Clusters 2–10 were named beginning from the one with the most members. Different clusters are indicated, and PAM sequences recognized by members of each cluster are highlighted with the associated color. The Cas9 which belongs to the same clade is outlined by a black dashed line. Sequences having known structures are marked red; their PDB code is shown in parentheses.

(Fig. 4b). Additionally, one ortholog, Ssa, retained 95% of its cleavage activity at 10 °C (Fig. 4a). We also observed that five Cas9 orthologs (Cme2, Cme4, Nsp, Khu, and Fma) retained <25% activity at reaction temperatures of 25 °C or below.

**Target DNA cleavage by Cas9 orthologs**. To characterize the termini resulting from Cas9 DNA cleavage, we developed a method that allows both termini resulting from target cleavage to be captured simultaneously in a deep-sequencing read (Supplementary Fig. 10). To validate the approach, we examined the cleavage positions for restriction endonucleases HhaI, FspI, and HinP1I, and for Spy and Sau Cas9. As shown in Fig. 5a, restriction enzymes generated defined cut-sites, either blunt-ended or staggered (5′- or 3′-end overhangs) as expected. This can be contrasted with Spy Cas9 target cleavage that, depending on the target site, generated either blunt-end cuts, 1-nt 5′-staggered termini due to RuvC post-cleavage trimming, as observed previously[40], or a mixture of both (Supplementary Fig. 11). In all, as averaged across five targets, Spy Cas9 generated predominantly blunt-ended DNA target cleavage, as reported previously[4] (Fig. 5b). Analysis of Sau Cas9 target cleavage produced almost entirely blunt-ended products as shown earlier[40] (Fig. 5b and Supplementary Fig. 11 and Supplementary Data 5).

We next evaluated the target cleavage pattern for 19 orthologs from our collection. Some Cas9s produced blunt-ended termini like Sau Cas9 (e.g., Cpe and Tsp (Fig. 4c)) while others, depending on the target site, produced either blunt-ended or 1 nt 5′-overhang termini similarly to Spy Cas9 (e.g., Sag1 and

Seq1 (Supplementary Data 5)). Some orthologs, as averaged across five different target sites, consistently generated overhanging termini varying between one or more nts (e.g., Khu, Lpn, Nsa, and Esp1) (Fig. 5c and Supplementary Data 5). In these cases, only 5′-staggered-end-cleavage products were recovered with the non-target strand tending to terminate at multiple positions, suggesting variation in the positioning of or post-cleavage trimming by the RuvC domain while the target strand was cleaved predominantly between the 3rd and 4th positions of the protospacer (Fig. 5c and Supplementary Data 5). It should be noted that 5′ overhanging target cleavage (1 nt) was previously reported for a single type II-B Cas9 from *Francisella novicida* (Fno)[41] and the type II-B Cas9 characterized in our study, Lpn, exhibited a nearly identical cleavage pattern.

**Discussion**

We identified Cas9s with G-, C-, A-, and T-rich PAM recognition of varying compositions, altogether, greatly expanding the sequence space targetable by Cas9. The observed diversity in PAM length was also striking with the majority of orthologs recognizing PAMs greater than 2 bp. This difference may be important for genome editing applications as orthologs with longer PAM recognition (≥3 bp) may afford higher specificity[12,17,42]. Additionally, phylogenetic and clustering analyses revealed that the PI domain was not always congruent with the rest of the protein. For example, conservation of the PI domain among related Spy Cas9 proteins was 1.4 times lower relative to the N-terminal portion (Supplementary Fig. 7). In

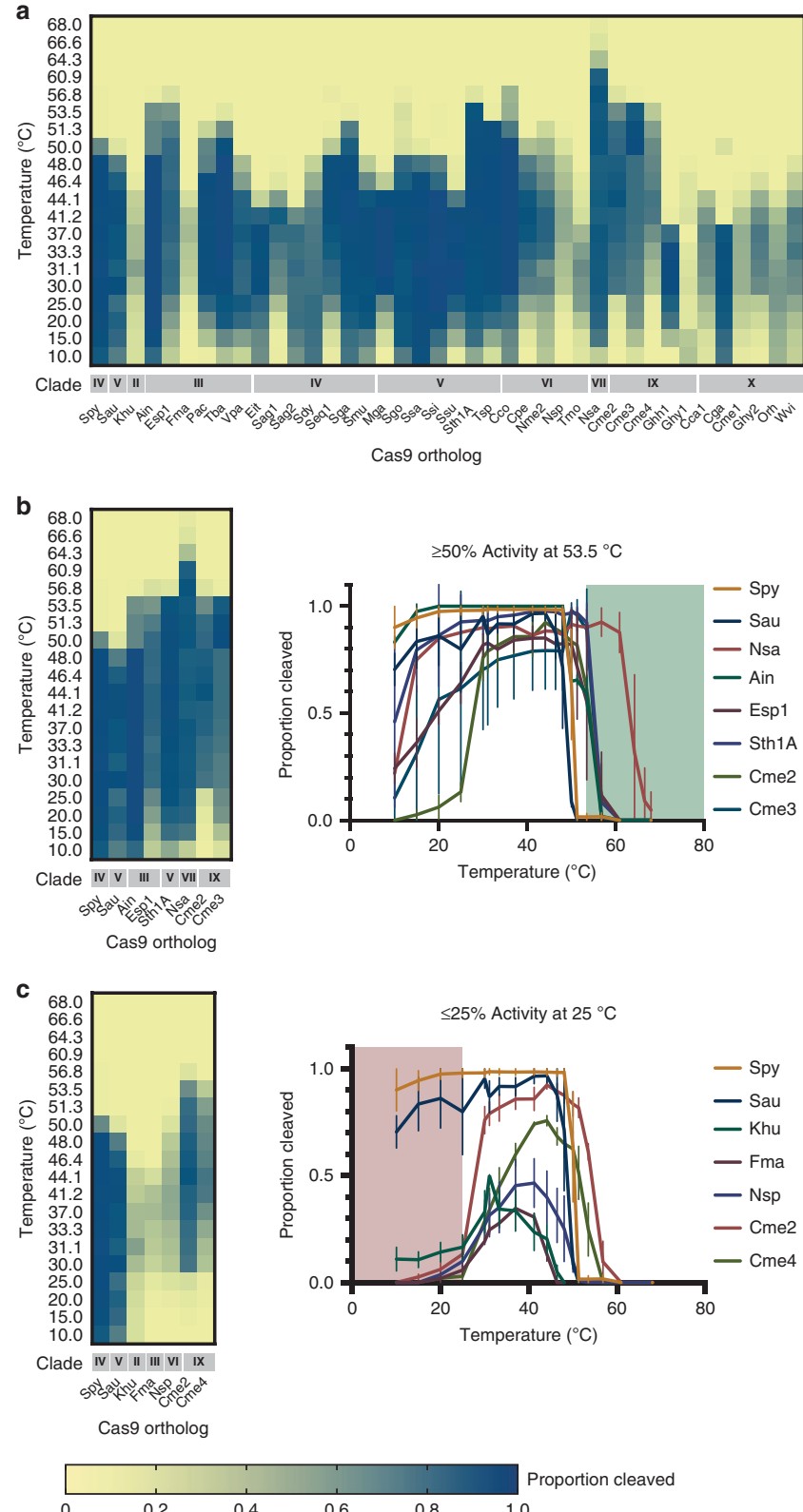

some cases, these differences were even greater as was noted for orthologs from *Neisseria meningitidis* (Nme) where PI domains only shared 52% identical residues while the rest of the protein was nearly the same (98% identity)[43] or in clade X where sequences belonged to three different homology clusters (Fig. 3; clusters 7, 5, and 10). Moreover, sequence variation from clusters 1, 3, and 4 when compared with the structures of Spy (4zto), Nme

(6kc8), Sau (5axw), and Sth (6rjd) could be modeled into just a single PI domain architecture (Supplementary Fig. 12). Altogether, these observations could in part be explained by the uncoupling of PI domain evolution from the rest of the protein, indicating that it is under selective pressure to diversify perhaps in response to PAM-based phage escape strategies, as described previously[44–46]. Additionally, they suggest that the Cas9 PI

**Fig. 4 Activity of Cas9 orthologs at varying temperatures.** The cleavage activity of Cas9 orthologs was measured using in vitro DNA cleavage assays using fluorophore-labeled double-stranded DNA (dsDNA) substrates. Cleaved fragments were quantitated and are represented in a heatmap **a** showing overall activity at temperatures ranging from 10 °C to 68 °C. The intensity of the blue color indicates the proportion of substrate cleaved. Source data are provided in the Source Data file. **b** Cas9 orthologs with activity at elevated temperatures. In vitro DNA cleavage activity for a subset of Cas9 orthologs with >50% activity at 53 °C is summarized in a heatmap and plotted as the proportion of DNA substrate cleaved at varied temperatures. The intensity of the blue color in heatmaps indicates the proportion of substrate cleaved. Points represent the mean ± SEM of at least three independent experiments. Green shading highlights the temperature range above 53 °C. **c** Cas9 orthologs with reduced activity at room temperature. In vitro DNA cleavage activity for a subset of Cas9 orthologs with <25% activity at 25 °C is summarized in a heatmap and plotted as a proportion of DNA substrate cleaved at varied temperatures. The intensity of the blue color in heatmaps indicates the proportion of substrate cleaved. Red shading highlights the temperature range below 25 °C Points represent the mean ± SEM of at least three independent experiments.

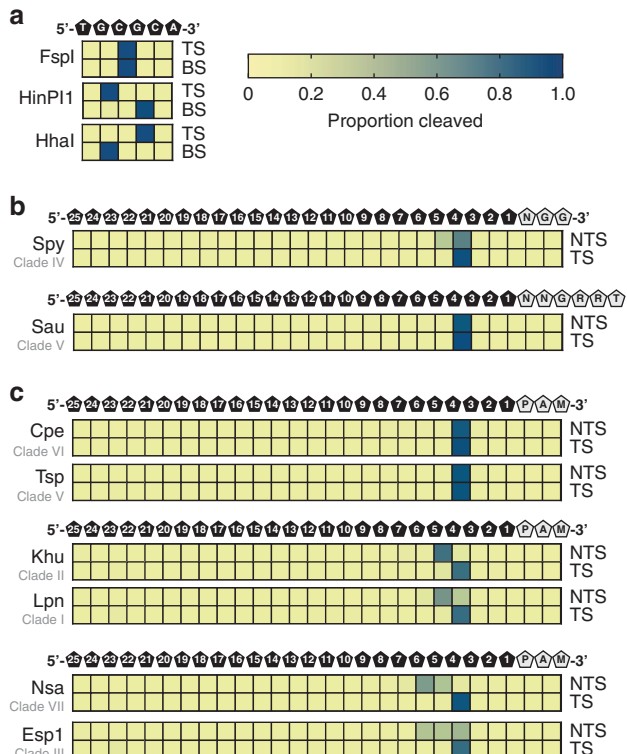

**Fig. 5 Target DNA cleavage patterns produced by Cas9 orthologs.** Cleavage sites and resultant double-stranded DNA (dsDNA) ends are depicted as heatmaps that show the proportion of cleaved ends recovered by DNA sequencing at each position of a target DNA. The intensity of the blue color indicates the proportion of mapped cleavage ends. **a** Control digests using restriction enzymes showed that blunt ends, 5'-overhangs and 3'-overhangs might be recovered with our approach. TS indicates the top strand; BS indicates the bottom strand. **b** Spy Cas9 and Sau Cas9 cleaved DNA ends. Heatmaps represent mapped cleavage ends as the averages at each position in five different dsDNA targets. The position of the DNA bases and protospacer adjacent motif (PAM) sequences is depicted above the heatmaps. NTS indicates a non-target strand; TS indicates the target strand. **c** Blunt and staggered-end cleavage. Examples of blunt, one base 5'-overhang staggered cleavage, and multiple base 5'-overhang cleavage are depicted as heatmaps that show the proportion of cleaved ends as the averages at each position in five different dsDNA targets. The position of the DNA bases and PAM sequences is depicted above the heatmaps. NTS indicates a non-target strand; TS indicates the target strand. Source data are provided in Supplementary Data 5.

domain is extraordinarily flexible and can be engineered to recognize a wide variety of sequence motifs encompassing the full spectrum of DNA nucleotides.

The gRNAs from our collection were in general conserved between related Cas9 proteins, although diverse gRNA structures

not previously observed were identified. In general, they could be classified into seven groups based on tracrRNA sequence and structural homology and visual inspection of sgRNA modules, as exemplified by Seq1 (Spy-like), Cca1, Tde, Sth1A, Lsp1, Nme2, and Gsp (Fig. 2 and Supplementary Fig. S3). Altogether, this may warrant the expansion of the number of discrete non-cross reactive Cas9 and sgRNA combinations from four to seven or more pending future studies[20,26]. This finding is important for orthogonal genome editing approaches where simultaneous, yet disparate activities are required at different sites[20,47].

Finally, an in-depth evaluation of DNA cleavage activity of the Cas9 nucleases described here exposed additional differences among orthologs. These included a wide range of temperature dependencies. Of particular interest was Cme2 Cas9, which was only robustly active from ~30 °C to 55 °C suggesting the possibility of temperature-controlled DNA search and modification. Additionally, the DNA cleavage activity at different temperatures for Nsa and Ssa Cas9s suggested they could be harnessed for use in thermo- or psychrophiles, respectively. Furthermore, we characterized orthologous Cas9 nucleases with different and potentially advantageous properties compared to those generally prescribed to Spy Cas9. These included variation in the termini resulting from target cleavage as well as a preference for a longer tract of gRNA and DNA target-site homology.

## Methods

**Identification and phylogeny of Cas9 orthologs.** Type II Cas9 endonucleases were identified by searching for the presence of an array of CRISPRs using PILER-CR 1.06[48]. Following identification, the DNA sequences surrounding the CRISPR array (about 15 kb 5' and 3' of the CRISPR array) were examined for the presence of open-reading frames (ORFs) encoding proteins >750 amino acids. Next, to identify *cas* genes encoding Cas9 orthologs, multiple sequence alignment of sequences from a diverse collection of Cas9 proteins was performed using MUSCLE 3.8.31[49] and then used to build profile hidden Markov models (HMMs) for Cas9 sub-families using HMMER 3.2.1[50]. The resulting HMMs were then utilized to search protein sequences translated from the *cas* ORFs for the presence of genes with homology to Cas9. Alternatively, Cas9 orthologs and the metagenomic sequence encoding them were obtained from publicly available datasets through the Joint Genome Institute's Integrated Microbial Genomes & Metagenomes resource (IMG/M): https://img.jgi.doe.gov/cgi-bin/m/main.cgi [51]. Only proteins containing the key HNH and RuvC nucleolytic domains and catalytic residues defining a type II Cas9 protein[52] were selected (Supplementary Data 6). Through phylogenetic analyses (MEGA7 10.0.5[53]), Cas9 proteins were then parsed into distinct families and representative members of each group used to select orthologs for characterization. To place our collection in context with previously described Cas9 orthologs, a phylogenetic tree was built using type II-A, -B, and -C representatives[34] and those we selected for characterization using MEGA7[53] employing Neighbor-Joining[54] and Poisson correction[55] methods.

**Engineering single-guide RNA solutions.** The trans-activating CRISPR RNA (tracrRNA) essential for CRISPR RNA (crRNA) maturation[56] and Cas9-directed target-site cleavage in type II systems[4,57] was identified by searching for a region in the vicinity of the *cas9* gene, the anti-repeat, which may base-pair with the CRISPR repeat and was distinct from the CRISPR array(s). Once identified, the possible transcriptional directions of the putative tracrRNAs for each system were established by examining the secondary structures using UNAfold 3.9[58] and possible termination signals present in RNA versions corresponding to the sense and anti-sense transcription scenarios surrounding the anti-repeat. Based on the likely transcriptional direction of the tracrRNA and CRISPR array, single-guide RNAs

(sgRNAs), representing a fusion of the CRISPR RNA (crRNA) and tracrRNA[4], were designed. For each ortholog, this was accomplished by linking 16 nt of the crRNA repeat to the complementary sequence of the tracrRNA anti-repeat by a 4 nt GAAA loop similar to that described previously for Spy Cas9[4]. All repeat, tracrRNA sequences, and sgRNA solutions are listed in Supplementary Data 2.

**Computational analysis of Cas9 tracrRNAs.** BLAST 2.7.3 (with parameters to optimize finding short sequences in highly repetitive regions (-task blastn_short -dust no))[59] was used to identify sequences homologous to the 79 identified tracrRNAs. The resulting collection of identified sequences were grouped using CD-HIT 4.7[60] at a 90% sequence similarity threshold. The resulting clusters were filtered to remove groups that did not contain at least one of the 79 reference tracrRNA sequences. Next, sequence homology and secondary structure models were constructed for each group using MAFFT 7.407[61] and RNAalifold 2.4.5[62], respectively. Both models were then used to search for sequence/structural homology in the full set of reference and BLAST-identified sequences using the RNA structure search tools in the Infernal 1.1 software suite[63]. The structural overlap between clusters was then generated by comparing the results of each covariance model (CM). To graph the relationship among tracrRNAs, vertices were first added for each representative CM (sequences with both shared secondary structure predictions and at least 90% sequence similarity). If two vertices shared a CM, they were connected with a line weighted by the percent similarity between shared vertices (percent similarity = (# of shared sequences)/(min(# found by model 1, # found by model 2))).

**Production of sgRNAs.** All sgRNA molecules used in this study were synthesized by in vitro transcription using HiScribe™ T7 Quick High Yield RNA Synthesis Kits (New England Biolabs), or transcribed directly in the in vitro translation (IVT) reaction. Templates for sgRNA transcription were generated by PCR amplifying synthesized fragments (IDT and Genscript) or by annealing a T7 primer oligo to a single-stranded template oligonucleotide. Transcribed RNA products were treated with DNaseI (New England Biolabs) to remove DNA templates and cleaned up with Monarch RNA Cleanup Kit (50 μg) (New England Biolabs) and eluted in nuclease-free water. RNA concentration and purity were measured by NanoDrop spectrophotometry, and RNA integrity was visualized by SYBR™ Gold staining of reaction products separated on Novex TBE-Urea 15% denaturing polyacrylamide gels with 0.5× TBE (Tris borate EDTA) buffer.

**PAM library cleavage using in vitro translation.** Cas9 was produced by IVT using either a continuous exchange 1-Step Human Coupled IVT Kit (Thermo Fisher Scientific) or a PURExpress bacterial IVT kit (New England Biolabs), following the manufacturer's recommended protocol similar to that described previously[33]. Plasmid DNA encoding human or E. coli codon optimized Cas9s were generated for use as templates for IVT reactions. Synthetic DNA fragments were synthesized by Genscript, Inc. and Twist Bioscience and assembled by NEBuilder HiFi DNA Assembly kit (New England Biolabs) into pT7-N-His-GST (Thermo Fisher Scientific) or pET28a (EMD Millipore). Following IVT, 20 μl of supernatant containing soluble Cas9 protein was mixed with RiboLock RNase Inhibitor (40 U; Thermo Fisher Scientific), and 2 μg of T7 in vitro transcribed sgRNA and incubated for 15 min at room temperature. Alternatively, the sgRNA was transcribed directly in the IVT kit by supplying a DNA template containing a T7 promoter and sequence encoding the respective sgRNA. In this situation, 0.5 μg of plasmid encoding the *cas9* gene and a 100-fold molar excess of sgRNA template were added to the IVT reaction mix. In all, 10 μl (or series of tenfold dilutions) of the resulting Cas9-sgRNA ribonucleoprotein (RNP) complex were then combined with 1 μg of the 7 bp randomized PAM library described previously[19] in a 100 μl reaction buffer (10 mM Tris-HCl pH 7.5 at 37 °C, 100 mM NaCl, 10 mM MgCl$_2$, 1 mM DTT) and incubated for 60 min at 37 °C.

**Capture and sequencing of cleaved library fragments.** Cleaved library fragments were captured by adapter ligation, enriched for by PCR amplification, and deep sequenced as described earlier[19]. Briefly, cleaved libraries were first subjected to DNA end-repair by incubation with 0.3 μl (1U) of T4 DNA polymerase (New England Biolabs) and 0.3 μl of 10 mM dNTP mix (Thermo Fisher Scientific) for 15 min at 12 °C and inactivated by heating (75 °C for 20 min). To efficiently capture free DNA ends, a 3'-dA overhang was added by incubating the reaction mixture with 0.3 μl (1.5 U) of DreamTaq polymerase (Thermo Fisher Scientific) for 30 min at 72 °C. The resulting DNA was then purified (Monarch PCR & DNA Cleanup purification column (New England Biolabs)) and ligated to adapters with a 3' dT overhang with 1 μl 400 U of T4 Ligase (New England Biolabs) in 25 μl of ligation buffer (50 mM Tris-HCl, pH 7.5 at 25 °C, 10 mM MgCl$_2$, 10 mM DTT, 1 mM ATP, 5% (w/v) PEG 4000). After 1 h at room temperature, 10 μl of the ligation reaction was used as the template in a PCR reaction (Q5 DNA polymerase (New England Biolabs); 15 cycles; 100 μL of final reaction volume) containing primers specific to the PAM-side of the library and the adapter. DNA was next purified (Monarch PCR & DNA Cleanup purification column (New England Biolabs)) and the sequences and indexes required for Illumina deep sequencing were incorporated through two rounds of PCR (Phusion High-Fidelity PCR Master Mix in HF buffer (New England Biolabs); ten cycles each round; 50 μL of final reaction

volume). The resulting products were then deep sequenced on a MiSeq Personal Sequencer (Illumina) with a 25% (v/v) spike of PhiX control v3 (Illumina).

**Identification of PAM preferences.** PAM sequences that supported dsDNA target cleavage were determined as described earlier[19,33,64]. Briefly, after sequencing, the location of cleavage within the library protospacer was first assessed by evaluating the position with the greatest number of adapter-ligated reads using a custom script[65]. The PAMs associated with library fragments that supported cleavage were then extracted[65] and used to evaluate the bias in the bp composition at each position within the randomized PAM library relative to that in the starting library by normalization ((treatment frequency)/((control frequency)/(average control frequency))). Next, PAM preferences were quantified using position frequency matrices (PFMs) and displayed as a WebLogo. Analyses were limited to the top 10% most frequent PAMs to reduce the impact of background noise resulting from non-specific cleavage coming from other components in the IVT mixtures.

**Computational analysis of Cas9 PAM interacting domains.** The Cas9 orthologs characterized here were aligned using MAFFT 7.407[61]. Their PAM interacting (PI) regions corresponding to the C-terminal domain of *Streptococcus pyogenes* Cas9 (4ZT0_A:1090-1365) were extracted and used as queries for two iterations of PSI-BLAST 2.2.26[66] search against the NCBI NR protein collection, UniRef100 and MGnify[67] databases. Hits were extracted, filtered to 80% identity using CD-HIT 4.6[60] and clustered with CLANS 1.0[68]. CLANS is an implementation of the Fruchterman–Reingold force-directed layout algorithm, which treats protein sequences as point masses in a virtual multidimensional space, in which they attract or repel each other based on the strength of their pairwise similarities (CLANS *P* values). CLANS *P* values are calculated from BLAST *E*-values by dividing them by effective search space used. Resulting CLANS networks were visually inspected, and clusters were identified. For groups larger than 150 sequences (Supplementary Data 4), a phylogenetic analysis was performed recovering sequences that were filtered out during the previous step and removing identical ones. Next, multiple sequence alignments were performed for clusters 1–6 using MAFFT (options: "–ep 0.123–maxiterate 20–localpair") and regions with gaps removed with trimAL 1.2[69] (option: "-gt 0.01"). Lengths of the resulting alignments varied from 359 to 652 residues in clusters 2 and 3, respectively. Phylogenetic trees were generated using IQtree 1.6.10[70] with auto model selection and 1000 fast bootstrap (options: "-alrt 1000 -bb 1000").

**Cas9 expression and purification.** Spy, *S. thermophilus* CRISPR3 (Sth3), and *S. thermophilus* CRISPR1 (Sth1) Cas9 proteins cloned into the pBAD-Chis vector[19] were expressed in *E. coli* DH10B strain at 16 °C for 20 h in the presence of 0.2% (w/v) arabinose. Other orthologs were first *E. coli* codon optimized and cloned into the pET28 vector yielding constructs encoding fusion proteins comprising a C-terminal 6-His-tag. In some instances, sequences encoding nuclear localization sequences (SV40 origin) were incorporated onto the 5′ and 3′ ends of the *cas9* gene. The expression of each ortholog was then tested in different *E. coli* strains (NiCo21(DE3), T7 Express lysY/Iq, NEB® Express Iq) under various growth conditions (media, temperature, induction) with the amount of protein produced being measured by SDS-PAGE analysis. Optimized conditions were then chosen for flask scale purification. Cells were disrupted by sonication. The supernatant was loaded onto HiTrap DEAE Sepharose (GE Healthcare), followed by subsequent purification on Ni2 +-charged HiTrap chelating HP column (GE Healthcare) and HiTrap Heparin HP (GE Healthcare) columns. Purified Cas9 proteins were stored at −20 °C in 20 mM Tris-HCl, pH 7.5, 500 mM KCl, 1 mM EDTA, 1 mM DTT, and 50% (v/v) glycerol.

**Evaluation of protospacer cleavage patterns.** To capture protospacer cleavage patterns with single-molecule resolution, we developed a minicircle double-stranded (ds) DNA substrate that allows both ends of target cleavage to be captured in a single Illumina sequence read. First, 124 nt oligonucleotides (IDT) (see Supplementary Data 7) were circularized using with CircLigase™ single-stranded (ss) DNA Ligase (Lucigen) according to the manufacturer's suggestion. Circularized ssDNA was next purified and concentrated using a Monarch® PCR & DNA Cleanup Kit (NEB). In total, 20 pmol of the purified product was then incubated with 25 pmol of a complementary primer in 1× T4 DNA ligase buffer (NEB) supplemented with 40 μM dNTPs. To allow the primer to anneal, the reaction was then heated to 65 °C for 30 s followed by a decrease in temperature to 25 °C at a rate of 0.2 °C/s. Six units of T4 DNA polymerase and 400 units of T4 DNA ligase (NEB) were then added, and the reaction was incubated at 12 °C for 1 h to allow second strand synthesis. Following purification with a Monarch® PCR & DNA Cleanup Kit and elution into 1X CutSmart® buffer (NEB) containing 1 mM ATP, 15 units of Exonuclease V (RecBCD; NEB) and T5 exonuclease (NEB) were added to the sample and incubated at 37 °C for 45 min. In total, 0.04 units of proteinase K (NEB) were then added, and the sample was incubated at 25 °C for 15 min prior to purification with a Monarch® PCR & DNA Cleanup Kit. After elution, the yield of circular dsDNA was assessed using an Agilent 2100 Bioanalyzer.

For minicircle digestion, Cas9 RNPs were formed by incubating 1 pmol of sgRNA with 0.5 pmol of Cas9 protein in 1× NEBuffer™ 3.1 or 2.1 (NEB) at room temperature for 10 min. In all, 0.1 pmol of circular dsDNA substrate was added, samples were incubated for 15 min at 37 °C, and then each 20 μl reaction was

quenched by the addition of 5 µl of 0.16 M EDTA. Reactions were concentrated and purified with a Monarch® PCR & DNA Cleanup Kit, and the entire 8 µl of eluted product was used as a substrate for Illumina sequencing library construction using a NEBNext® Ultra™ II DNA Library Prep Kit for Illumina® (NEB) and the protocol provided with the kit. Fifteen cycles of PCR were used to add the Illumina priming sequences and index barcodes, and then the concentration of each reaction was assessed on an Agilent 2100 Bioanalyzer. Libraries were pooled and sequenced on either an Illumina NovaSeq or NextSeq instrument with 2 × 150 paired-end sequencing runs. Cleavage sites were then mapped using custom scripts[71] and visualized as heatmaps (representing proportion cleaved) using Microsoft Excel 16.36 and GraphPad Prism 8.

**In vitro cleavage assays for determining optimal buffer, temperature, and spacer length.** First, DNA substrates containing a canonical PAM for each ortholog were amplified from HEK293T genomic DNA by PCR using primers corresponding to WTAP and RUNX1. Forward primers were labeled with 5′-FAM and 5′-ROX for WTAP and RUNX1, respectively. Reverse primers were unlabeled. In total, 515 and 605 bp PCR products for WTAP and RUNX1, respectively, were then purified with a Monarch® PCR & DNA Cleanup Kit (5 µg) (NEB T1030S) and DNA concentration and purity measured by NanoDrop™ spectrophotometry (Thermo Fisher). Purified Cas9 protein was then diluted to 1 µM in dilution buffer (300 mM NaCl, 20 mM Tris, pH 7.5) and stored on ice. Next, sgRNAs (Supplementary Data 2 and 7) were diluted to 2 µM in nuclease-free water. Cas9 and sgRNA were then combined in a 2:1 sgRNA: Cas9 molar ratio in reaction buffer at room temperature for 10 min. The substrate was added next at a Cas9:sgRNA:DNA ratio of 10:20:1 and incubated for 30 min. For buffer optimization and spacer length preference experiments, 1× NEBuffers 1.1, 2.1, 3.1, or CutSmart (NEB B7200S) were used as reaction buffers, and incubations took place at 37 °C. For thermoactivity experiments, reactions were performed in NE buffer 3.1. Here, RNPs were initially formed at room temperature and then transferred to a thermal cycler pre-heated or cooled to the various assay temperatures prior to DNA substrate addition. 10× DNA substrate (100 nM) was separately equilibrated at the designated temperature prior to being added to the RNP containing reaction tubes. Reactions were quenched by adding SDS to 0.8% (v/v) and 80 mU Proteinase K (NEB P8107S). Cleavage products were diluted 4× in nuclease-free water and subjected to capillary electrophoresis (CE) to quantify the extent of cleavage[72]. The fraction of substrate cleaved at each temperature was then visualized as heatmaps, using Microsoft Excel 16.36 and GraphPad Prism 8.

**Cas9 protein thermal stability.** Purified Cas9 proteins were diluted in 300 mM NaCl, 20 mM Tris, pH 7.5 to 5–10 µM at room temperature. In total, 10 µL of the diluted protein was loaded into NanoDSF Grade Standard Capillaries (Nano-Temper), and melting temperatures were determined using a Prometheus NT4.8 NanoDSF instrument according to the manufacturer's instruction. The temperature was increased from 20 °C to 80 °C at a rate of 1 °C/min. Inflection points of melting curves are reported as the Tm.

**Reporting summary.** Further information on research design is available in the Nature Research Reporting Summary linked to this article.

## Data availability

Raw deep-sequencing data that support PAM and cleavage pattern determination for Cas9 orthologs are deposited in the NCBI Sequence Read Archive under BioProject IDs PRJNA631559 and PRJNA622541. All other relevant data are available from the corresponding authors on reasonable request. All protein sequences used for computational analysis are available in public databases (e.g., UniRef100, MGnify, IMG/M, PDB), full list of accession numbers and sequences are provided in Supplementary Data 4 and 6. Source data are provided with this paper.

## Code availability

Scripts used to analyze deep-sequencing data are available on GitHub[65,71].

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

## Acknowledgements

We thank Migle Stitilyte from CasZyme for the preparation of the sgRNA templates.

## Author contributions

G.G., J.K.Y., M.C., C.A., N.D.C., W.E.J., E.S., G.B.R., and V.S. designed the research; G.G., J.K.Y., T.K., D.K., T.U., M.J., M.M.G., S.P., P.W., Z.H., S.K.D., G.B., J.L.C., M.M., Z.S., R.T.F., and P.R.W. performed the research, and G.G., J.K.Y., C.A., N.D.C., W.E.J., E.S., G. B.R., C.V., and V.S. analyzed the data. G.G., J.K.Y., G.B.R., and V.S. wrote the paper. All authors read and approved the final paper.

## Competing interests

Z.H., J.K.Y., G.G., and V.S. have filed patent applications related to the paper. G.G, T.U, M.J., and M.M.G. are employees of CasZyme. J.K.Y., S.P., Z.H., C.A., and N.D.C. are employees of Corteva Agriscience. J.L.C., M.M., R.T.F, E.S., P.R.W., Z.S., W.E.J., and G.B.R. are employees of NEB. V.S. is a Chairman of CasZyme. V.S. and G.G. have a financial interest in CasZyme. The remaining authors declare no competing interests.
