## [Peer Review File · Nature Communications]

REVIEWER COMMENTS

Reviewer #1 (Remarks to the Author):

Gasiunas, Young et al. describe a tour-de-force effort to characterize 79 Cas9 orthologs. Although a huge variety of orthologs have been identified bioinformatically, mechanistic and genome editing studies have only focused on a handful of Cas9's. Here, the authors selected a large and diverse group of orthologs for characterization. In particular, they use, to impressive effect, a previously established in vitro transcription/translation system to characterize PAM requirements for these orthologs. In addition to a significant amount of interesting bioinformatic analysis (e.g. sgRNA requirement, correlating PAM-interacting domain homology with PAM requirements), the authors also performed in-depth biochemical characterization of a large subset of the orthologs. They develop a new method using DNA minicircles to map cut sites for these orthologs, a method that is likely to be widely useful to the community. In addition, they characterize optimal buffer requirements, spacer length and Cas9 thermostability.

This is an exceptionally well-written and easy to follow manuscript. The data provided in the paper will be a huge asset to the CRISPR and genome editing communities. I only have a few suggestions on clarifications to text and figures, but otherwise am supportive of accepting the manuscript as is.

Minor comments:

1. In the Results section, the authors state that IVT RNP mixtures were diluted from 10^1 to 10^3 in 10-fold increments and tested for cleavage activity. However, in Fig. S4A, dilutions of 10^5 or 10^6 are shown for the three orthologs that have been described previously (Spy, Sth3 and Sth1). Some clarification could be added to either the Results or the Figure Legend for Fig. S4A. Were the new orthologs tested in this study simply less active, requiring PAM identification at higher concentrations?
2. Last sentence of first paragraph of Introduction: While I agree that only a handful of Cas9 variants have been used for genome editing, the citations in this sentence are not comprehensive. For example, the current reference 60 could also be added to this sentence.
3. Please add the color code for the various regions of sgRNAs shown in Fig. S3 to the figure legend (e.g. repeat-antirepeat duplex: green, nexus: purple and 3'-hairpin: blue).
4. It is unclear why some bars are colored differently in Fig. S9.
5. Just a suggestion and not a requirement: I wonder if it would be more intuitive to look at Figures S9 and 4A if the Cas9 orthologs shown in ascending order of thermal stability. The authors could leave Spy and Sau as the first two shown, as those are benchmark Cas9 orthologs. The remaining orthologs could be ordered from lowest to highest thermal stability. This may make it easier to follow the data in Fig. 4A, as we should see a clear trend upward in the optimal temperatures for cleavage. However, the authors do draw attention to the similar orthologs in Fig. 4B-C, so this change may not be necessary.
6. For Fig. S10, Cas9 cuts DNA cut within the target site, so it seems there should be red on either side of the linear DNA following cleavage. My understanding of the method is that the cut site is determined by looking at both the R1 and R2 reads, so depicting target regions on either end of the sequenced DNA is an important detail for understanding the method.

Reviewer #2 (Remarks to the Author):

Cas9 has become the most popular genome editing tool in recent years. While important for efficient target searching and high fidelity, the PAM restriction of each Cas9 limits the targetable sequence space. It is highly desirable to have a catalog of well-characterized Cas9 enzymes, each with different PAM specificity, so that researchers can pick and choose for their specific genome editing needs. This study serves this exact purpose by sparse sampling the Cas9 sequence space and define the key parameters for each homolog. These include the carefully defined PAM code, the crRNA guide length, the preferred temperature profile, and the cleavage pattern. This reviewer finds the data in this study of high standard and highly useful. Some of the technical methods are quite novel. The cleavage-based PAM code definition is more desirable than the eliminating-based PAM definition method in previous studies. The ssDNA based deep sequencing method to define the cleavage pattern of Cas9s will also gain popularity in later studies. One parameter that was noticeably missing in this study is the off-targeting profile of each Cas9. How do they compare with the existing ones, such as the SpCas9 and SauCas9? It would also be nice if the authors could show one or two new tricks these Cas9s could do. For example, whether some are more efficient at mediating editing in zebrafish (cooler than 37°C, hence SpCas9 does not work efficiently), or more efficient at mediating knock-ins. Given that there are enough novelty in this study, I don't think asking more is appropriate. The writing and figures are in good shape for publication.

Here are a few minor points for the authors to improve the manuscript:

1. Fig. S4B: Any data to correlate dilution fold with editing efficiency? This may help the researchers to decide which concentration to use for their editing experiment.
2. Fig. S10: Do I get it right that the cleavage pattern analysis in this study biases towards the 5'-protruding ends? Can it resolve a 3'-protruding cleavage pattern? If not, this should noted in the text.
3. Figure S11: Confusing as to where the 5' and 3' ends are because the NTS is displayed on the top. Consider labeling the 5' and 3' ends for the top set of cleavage pattern in each group.

Reviewer #3 (Remarks to the Author):

Gasiunas et al. describe the broadest analysis yet of Cas9 functional diversity. They classify numerous clades based on sequence alignments, retrieve predicted guide sequences from the native CRISPR-Cas loci, and then use in vitro translation to sample dozens of the Cas9/guide combinations and define their PAM requirements.

There are several aspects of this manuscript that will be appreciated by the field. The first, as noted above, is the sheer breadth of the sampling – far beyond what anyone else has done before. A second novel aspect of the work is the separate analysis of PAM-interacting domains, as this provided insight into separate domain evolution that has not been described previously elsewhere. Similarly, the analysis and classification of guide RNA sequence/structure adds considerable value to the existing literature. The main limitation of the current work is that it provides the reader with no indication whatsoever of which of these systems will actually prove useful for genome editing (especially in eukaryotic cells) and which will not, despite the fact that the introduction uses genome editing to justify this extensive analysis. It would be unrealistic to expect the authors to do this for 79 orthologs, but why was a subset not validated for editing? Especially those with novel PAMs relative to Cas9s already reported?

There are a number of other smaller points that should be addressed in future versions of this manuscript.

1. The authors do a poor job of pointing the non-expert reader to the full roster of Cas9s (and corresponding references) already analyzed and tested to date, especially for those that are validated for mammalian editing, or reported in the last 1-2 years, or both. Accuracy on this point is clearly relevant for the reader's complete view of the context of the current manuscript and the state of the field that it addresses. Supplementary Table 1 of PMID 32572269 provides a more complete line-up, with references, though even this list is already slightly out of date (e.g. it does not include the especially recent PMIDs 32226015 and 32424114).

2. Page 5, bottom; descriptions of spacer lengths. Why was natural crRNA length not used as an indicator of lengths to test in vitro? This has proven useful in the past, with native crRNA lengths correlating well (not surprisingly) with guide lengths that work well in heterologous contexts. The authors also state that "It was previously reported that *Sau* and *Geo* Cas9 proteins require a longer spacer to function robustly." It is implied but never properly stated that "longer than" refers to the 20nt length known for *Spy*Cas9. Also, this is a more common phenomenon than the authors imply, as it was also reported for *Cje* (28220790) and *Nme2* (30581144) Cas9s.

3. Staggered cleavage sites: were any of those exhibiting staggered sites Type II-B? Staggered sites are already known in these cases (28387220). This report should be noted and referenced.

4. Uncoupling of PI domain evolution from the rest of the protein: the authors state correctly that this likely reflects compensation for PAM mutations that evade interference. However, it could also reflect evasion of anti-CRISPR inhibition for those anti-CRISPRs (e.g. *AcrIIA4*) that recognize PI domains.

Point-by-point respond to reviewers' comments

Reviewer #1:

Minor comments:

1. In the Results section, the authors state that IVT RNP mixtures were diluted from 10^1 to 10^3 in 10-fold increments and tested for cleavage activity. However, in Fig. S4A, dilutions of 10^5 or 10^6 are shown for the three orthologs that have been described previously (Spy, Sth3 and Sth1). Some clarification could be added to either the Results or the Figure Legend for Fig. S4A. Were the new orthologs tested in this study simply less active, requiring PAM identification at higher concentrations?

Reply:

We don't believe that the new orthologs are less active than those described previously and have updated Fig. S4A to be consistent with the conditions used in examining PAM recognition for the new orthologs described in our manuscript. For this, we included weblogs from 10^3 dilutions (IVT) for Spy, Sth1 and Sth3. We hope this will improve the clarity of the figure and results presented.

2. Last sentence of first paragraph of Introduction: While I agree that only a handful of Cas9 variants have been used for genome editing, the citations in this sentence are not comprehensive. For example, the current reference 60 could also be added to this sentence.

Reply:

Additional citations included as suggested.

3. Please add the color code for the various regions of sgRNAs shown in Fig. S3 to the figure legend (e.g. repeat-antirepeat duplex: green, nexus: purple and 3'-hairpin: blue).

Reply:

The figure legend was modified accordingly.

4. It is unclear why some bars are colored differently in Fig. S9.

Reply:

We have removed the coloring from Fig. S9 and added clade designations for each ortholog.

5. Just a suggestion and not a requirement: I wonder if it would be more intuitive to look at Figures S9 and 4A if the Cas9 orthologs shown in ascending order of thermal stability. The authors could leave Spy and Sau as the first two shown, as those are benchmark Cas9 orthologs. The remaining orthologs could be ordered from lowest to highest thermal stability. This may make it easier to follow the data in Fig. 4A, as we should see a clear trend upward in the optimal temperatures for cleavage. However, the authors do draw attention to the similar orthologs in Fig. 4B-C, so this change may not be necessary.

Reply:

We thank the reviewer for this thoughtful suggestion. We have now edited Fig. 4A-C to add clade designations to the Cas9 orthologs shown. We hope that our edits based on this suggestion have

improved the clarity of the figure and highlight the lack of a clear relationship between temperature and cleavage activity between the clades.

6. For Fig. S10, Cas9 cuts DNA cut within the target site, so it seems there should be red on either side of the linear DNA following cleavage. My understanding of the method is that the cut site is determined by looking at both the R1 and R2 reads, so depicting target regions on either end of the sequenced DNA is an important detail for understanding the method.

Reply:

Indeed, as pointed out, there should have been red on either end of the cleaved DNA minicircle depicted in the workflow. We have edited the figure to include this.

Reviewer #2:

Here are a few minor points for the authors to improve the manuscript:

1. Fig. S4B: Any data to correlate dilution fold with editing efficiency? This may help the researchers to decide which concentration to use for their editing experiment.

Reply:

We don't think that the IVT fold dilution at which target cleavage activity was detected correlates with editing efficiency in cells as it is based on a biochemical assay not a cellular one. To improve the figure and clarify our results, we have updated Fig. S4A to be consistent with the conditions used in examining PAM recognition for the new orthologs described in our manuscript. For this, we included weblogs from 10^3 dilutions (IVT) for Spy, Sth1 and Sth3.

2. Fig. S10: Do I get it right that the cleavage pattern analysis in this study biases towards the 5'-protruding ends? Can it resolve a 3'-protruding cleavage pattern? If not, this should be noted in the text.

Reply:

The cleavage patterns that we observed for Cas9 orthologs were either blunt or 5'-overhanging ends. We have edited the results section of the manuscript to add a specific mention of this in the text. The method is able to resolve 3'-overhanging ends. This is shown in Figure 5A where the cleaved ends of DNA resulting from HhaI RE digestion. We have edited the results section of the manuscript to specifically mention this point. Changes can be found in the final paragraph of the results section

3. Figure S11: Confusing as to where the 5' and 3' ends are because the NTS is displayed on the top. Consider labeling the 5' and 3' ends for the top set of cleavage pattern in each group.

Reply:

We have indicated the 5' and 3' ends of the NTS on the sequences depicted above the heatmaps showing the cleavage sites.

Reviewer #3:

There are several aspects of this manuscript that will be appreciated by the field. The first, as noted above, is the sheer breadth of the sampling – far beyond what anyone else has done before. A second novel aspect of the work is the separate analysis of PAM-interacting domains, as this provided insight into separate domain evolution that has not been described previously elsewhere. Similarly, the analysis and classification of guide RNA sequence/structure adds considerable value to the existing literature. The main limitation of the current work is that it provides the reader with no indication whatsoever of which of these systems will actually prove useful for genome editing (especially in eukaryotic cells) and which will not, despite the fact that the introduction uses genome editing to justify this extensive analysis. It would be unrealistic to expect the authors to do this for 79 orthologs, but why was a subset not validated for editing? Especially those with novel PAMs relative to Cas9s already reported?

Reply:

Future studies will focus on developing the orthologs described in the manuscript for genome editing applications. The purpose of the current manuscript is to present the exceptional biochemical diversity afforded by Cas9 orthologs. Please note the authors believe the unique biochemical features identified may be useful in a wide array of applications including but not limited to genome editing (as stated in the introduction (e.g. “Easily reprogrammed to recognize new DNA sequences, it has been widely adopted for use in a multitude of applications to edit genomic DNA, modulate gene expression visualize genetic loci, or detect targets in vitro⁵⁻⁹.” and “Furthermore, the biochemical and physical characteristics of Cas9s routinely applied, producing predominantly blunt-end DNA target cleavage^{3,4}, slow substrate release²⁴, low frequency of recurrent target site cleavage²⁵, gRNA exchangeability²⁶, temperature dependence²⁷, and size²⁸ may also be unfavourable for its varied applications.”)). To further underscore this point, the abstract was changed to de-emphasize genome editing as the main justification for the analysis.

There are a number of other smaller points that should be addressed in future versions of this manuscript.

1. The authors do a poor job of pointing the non-expert reader to the full roster of Cas9s (and corresponding references) already analyzed and tested to date, especially for those that are validated for mammalian editing, or reported in the last 1-2 years, or both. Accuracy on this point is clearly relevant for the reader's complete view of the context of the current manuscript and the state of the field that it addresses. Supplementary Table 1 of PMID 32572269 provides a more complete line-up, with references, though even this list is already slightly out of date (e.g. it does not include the especially recent PMIDs 32226015 and 32424114).

Reply:

We have updated the main text and Supplementary Table S1 to include additional context on how the Cas9s characterized in our manuscript relate to previous ones. In a similar fashion, citations in the text have also been updated.

2. Page 5, bottom; descriptions of spacer lengths. Why was natural crRNA length not used as an

indicator of lengths to test in vitro? This has proven useful in the past, with native crRNA lengths correlating well (not surprisingly) with guide lengths that work well in heterologous contexts. The authors also state that “It was previously reported that *Sau* and *Geo* Cas9 proteins require a longer spacer to function robustly.” It is implied but never properly stated that “longer than” refers to the 20nt length known for *Spy*Cas9. Also, this is a more common phenomenon than the authors imply, as it was also reported for *Cje* (28220790) and *Nme2* (30581144) Cas9s.

Reply:

Our approach to guide RNA spacer length is one that is based on functional tests. Reasoning that previously characterized orthologs prefer between 20-24 nt length spacers, we tested spacer lengths of 20 and 24 nt (for most orthologs) to initially gauge the effect on double-stranded DNA target cleavage. Indeed, future biochemical and cellular experiments (outside the scope of the current manuscript given the large number of orthologs) aimed at precisely establishing the preferred spacer length will be performed. Additionally, we have included changes in the text to specify that “longer than” refers to longer than 20 nts and included additional citations for *Cje* and *Nme*.

3. Staggered cleavage sites: were any of those exhibiting staggered sites Type II-B? Staggered sites are already known in these cases (28387220). This report should be noted and referenced.

Reply:

We have updated the text accordingly. Please see last paragraph in the Results section.

4. Uncoupling of PI domain evolution from the rest of the protein: the authors state correctly that this likely reflects compensation for PAM mutations that evade interference. However, it could also reflect evasion of anti-CRISPR inhibition for those anti-CRISPRs (e.g. *AcrIIA4*) that recognize PI domains.

Reply:

We have included an additional citation in the Discussion section to capture both concepts as noted by the Reviewer.